# Autologous Platelet-Rich Plasma (PRP) for Treating Androgenetic Alopecia: A Novel Treatment Protocol Standardized on 2 Cases

**DOI:** 10.3390/jcm11247327

**Published:** 2022-12-09

**Authors:** Ana Maletic, Ivo Dumic-Cule, Petar Brlek, Rado Zic, Dragan Primorac

**Affiliations:** 1Polyclinic Maletic, 43500 Daruvar, Croatia; 2University North, 42000 Varaždin, Croatia; 3St. Catharine Special Hospital, 10000 Zagreb, Croatia; 4Department of Plastic Surgery, University Hospital Dubrava, 10040 Zagreb, Croatia; 5Faculty of Medicine, Josip Juraj Strossmayer University of Osijek, 31000 Osijek, Croatia; 6Medical School, University of Split, 21000 Split, Croatia; 7Faculty of Dental Medicine and Health, Josip Juraj Strossmayer University of Osijek, 31000 Osijek, Croatia; 8Medical School, University of Rijeka, 51000 Rijeka, Croatia; 9Medical School REGIOMED, 96450 Coburg, Germany; 10Eberly College of Science, The Pennsylvania State University, State College, PA 16801, USA; 11The Henry C. Lee College of Criminal Justice and Forensic Sciences, University of New Haven, West Haven, CT 06516, USA

**Keywords:** androgenetic alopecia, platelet-rich plasma, hair loss, hair follicle

## Abstract

Platelet-rich plasma (PRP) treatment has emerged in recent years as a valuable, effective, and affordable treatment for androgenetic alopecia. Androgenetic alopecia is the most common type of alopecia, affecting both men and women, and is characterized by diminished hair follicles mainly pronounced in the frontal region and vertex. A considerable variety of PRP treatment regimens have been described so far, but there is no consensus on the standardization of PRP preparation or administration protocol. Our study was conducted on two patients to test the efficacy of a new PRP application protocol of only two treatments by using a combination of a PRP collecting device and a conventional kit. Efficacy of treatment was assessed after a 6-month follow-up by artificial intelligence (AI)-driven software on microscopic images of treated regions. An average number of hairs, cumulative hair thickness, and the number of follicular units increased in the vertex region of both patients by 30/59%, 35/53%, and 14/48%, respectively. The novel treatment regimen showed significant effectiveness in only six months.

## 1. Introduction

Platelet-rich plasma (PRP) was introduced in the 1990s to treat chronic wounds with delayed healing [1]. During the last decade, it gained popularity as a useful therapy for many conditions, and spread across different specialties such as orthopedics, sports medicine, urology, gynecology, ophthalmology, and head and neck surgery, plastic surgery, and aesthetic medicine [2]. PRP effectiveness is based on a concentration of various growth factors and cytokines contained within the platelet fraction after autologous preparation of plasma from a patient’s whole blood.

Among different indications in aesthetic medicine, PRP found its place as a valuable therapy for hair restoration in patients with androgenic alopecia [3]. Androgenetic alopecia is the most common cause of hair loss, affecting nearly 50% of the population during their lifetime [4].

Despite extensive utilization of PRP for this indication, preparation procedures and administration regimens of PRP have not yet been standardized [5]. A recently published review revealed a significant variation between different PRP preparation methods described in several studies [6]. More than 50% of studies included in the review utilized different PRP processing systems, and some of them are not even reproducible due to a lack of protocol details such as the centrifugation procedure or the platelet activation method. Treatment regimen (volume of PRP per treatment, number of treatments, number of injections) is still not defined by uniformly agreed guidelines [7]. According to available literature, 3–6 treatments were applied to patients with androgenetic alopecia in studies with satisfactory outcomes [8].

In this paper, we present two male patients, 57 and 34 years old, with androgenetic alopecia who underwent only two PRP treatments according to a novel protocol. In addition, we utilized novel AI software for the initial assessment and patient follow-up.

## 2. Materials and Methods

### 2.1. Patient Data

We enrolled two Caucasian male patients with androgenetic alopecia in this pilot study. Both patients were otherwise healthy and in good physical condition. Androgenetic alopecia was diagnosed and described utilizing the Hamilton Norwood scale [9]. Patients did not receive any other treatment option for androgenetic alopecia before or during this study. PRP treatment was applied according to the novel regimen, with the first application differing between the two patients.

### 2.2. PRP Treatment Protocol

Patients were treated according to our novel protocol with only two treatment points. The first injection contained a large volume of concentrated PRP solution activated by calcium (Arthrex Angel System), and was followed by a second, small-volume booster without activation. Arthrex Angel System is innovative tool that achieves a high concentration of thrombocyte and accompanied minimal concentration of white blood cells.

#### 2.2.1. Patient #1

According to the Hamilton Norwood scale, the first patient, 57 years old, had degree IV of alopecia. Therefore, the treatment plan encompassed both the frontal and vertex area. Whole blood (120 mL) was collected from a peripheral vein using sodium citrate as an anticoagulant. Arthrex Angel system was used to prepare concentrated platelet-rich plasma (cPRP). The instrument hematocrit percentage (HTC%) level was set to 2%, and the final PRP volume was set to 8 mL. With the above settings, 3 mL of total cPRP was obtained. 3 mL of cPRP was then combined with 5 mL of platelet-poor plasma (PPP) to produce 8 mL of cPRP with a 5.17-fold increase in platelets and a decrease in the concentration of leukocytes (0.71-fold decrease) and neutrophils (0.66-fold increase) relative to peripheral blood. The treated area was anesthetized with lidocaine spray (Lidokain Belupo, Koprivnica, Croatia). The 8 mL of cPRP was immediately injected into the treatment area (21 × 11 cm; frontal area + vertex) subcutaneously and intradermally using a 1 mL syringe with a 0.4 × 13 mm needle, 27 gauge, with approximately one sting every 1 cm. Additionally, 5 mL of PPP was injected intradermally in designated areas using a 32-gauge needle. Immediately after the last intradermal treatment, 2 mL of PPP were massaged into the hair.

#### 2.2.2. Patient #2

According to the Hamilton Norwood scale, the second patient had degree III Vertex of alopecia. Thus, the treatment was focused on the vertex region. Whole blood (60 mL) was collected from a peripheral vein using sodium citrate as an anticoagulant. Arthrex Angel system was used to prepare cPRP. The instrument hematocrit percentage (HTC%) level was set to 5% and the final PRP volume was set to 4 mL. With the above settings, 4 mL of total cPRP was obtained with a 5.69-fold increase in platelets, 1.83-fold increase in leukocytes, and a decrease in the concentration of neutrophils (0.83-fold increase) relative to peripheral blood. The treated area was anesthetized with lidocaine spray (Lidokain Belupo, Koprivnica, Croatia). 4 mL of cPRP was immediately injected into the treatment area (9 × 8 cm, only vertex) subcutaneously and intracutaneously using a 1 mL syringe with a 0.4 × 13 mm needle, 27 gauge, with approximately one sting every 1 cm. Additionally, 5 mL of PPP was injected intradermally in designated areas using a 32-gauge needle. Immediately after the last intradermal treatment, 2 mL of PPP were massaged into the hair.

The second PRP treatment was administered to both patients four months after the initial treatment, and the protocol was the same for both. PRP was prepared after collecting 10 mL of peripheral whole blood in tubes coated with sodium citrate. Whole blood was further processed in a one-step centrifugation process. The pellet of platelets was reconstituted in plasma to a total volume of 4 mL, without the addition of a platelet activator. 4 mL of cPRP formed this way was immediately injected into the treatment area (21 × 11 cm in the first patient and 9 **×** 8 cm in the second patient) subcutaneously and intradermally using a 1 mL syringe with a 0.4 × 13 mm needle, 27 gauge, with approximately one sting every 1 cm.

### 2.3. Assessment of Treatment Efficacy

The outcome of the tested treatment regimens was assessed after six months on macroscopic images (global photography) and analysis performed by AI-powered software TrichoLAB (Fotofinder, Bad Birnbach, Germany). Macroscopic images allowed us to define the type of hair loss according to the Hamilton Norwood scale at the beginning of the study and enabled comparison with the final result [9]. Hair loss type and further follow-up points were assessed independently by two physicians with more than ten years of experience in trichology.

TrichoLAB is an AI-powered tool that combines standard microscopy with automatic digital image analysis, which results in a measurement of all the important hair parameters in situ. Examination by trichoscopy included the acquisition of four high-resolution images: frontal (midline), temporal (3 cm above the ear), occipital (across occipital protuberance) and vertex area. In the present study, we performed TrichoLAB analysis before the commencement of treatment (time point 0) and on four defined time points after the treatment (first, second, fifth, and sixth month). TrichoLAB system utilizes acquired images to derive the following parameters of hair on four analysed regions: average number of hairs, average hair shaft thickness, number of thin hairs, number of mid hairs, number of thick hairs, number of single follicular units, number of double follicular units, number of triple follicular units, cumulative hair thickness, number of follicular units and the derived Sinclair scale.

## 3. Results

The two patients with androgenetic alopecia were initially diagnosed by the Hamilton Norwood scale, according to which patient #1 had degree IV of hair loss and patient #2 had degree III-Vertex alopecia. Due to the more severe alopecia in the first patient, initial treatment comprised more PRP compared to the second patient. The second PRP application was the same for both patients. In the second patient, only vertex was treated. The applied PRP treatment regimen resulted in significant improvement in hair growth, as seen in macroscopic images (Figure 1). 

Further detailed analysis was performed using the TrichoLAB AI-powered software. The data analysis allowed a comparison of initial (before treatment) and final values (at 6-month follow-up), and the difference was expressed as a percentage.

An average number of hairs and number of follicles increased in the vertex region in the first patient six months after initial PRP administration. Cumulative hair thickness and average hair shaft thickness were enhanced. Two and three follicular hair units showed increase in number. In contrast, the number of one hair follicular unit and the number of thin hairs decreased (Figure 2). Treatment of the second patient showed similar trends, with specifically marked increase in number of hairs, number of follicles, cumulative hair thickness and three hair follicular unit (Figure 3).

PRP administration in frontal region of the first patient increased number of hairs and follicles, cumulative hair thickness, and both double and triple follicular units (Figure 4). Number of thin hairs and single follicular units decreased in size after therapy.

The average number of hairs, cumulative hair thickness, and the number of follicular units significantly increased in the vertex region of both patients. The same significant increase in the average number of hairs, cumulative hair thickness, and a number of follicular units was seen in the frontal area of the first patient. However, this effect was less profound than in the vertex area. The analysis of the occipital region of both patients showed a slight increase in main hair growth parameters. By contrast, no difference was recorded in the temporal regions of both patients. We can conclude that the novel treatment regimen using only two treatments displayed significant effectiveness within six months.

## 4. Discussion

The potential of autologous PRP to stimulate hair growth is well documented [10]. Based on recently published studies, we decided to combine two techniques of PRP preparation. Initially, we utilized Arthrex Angel System to collect PRP from the patients’ blood. The efficacy of this approach was recently tested against two commercially available systems [11]. After such treatment, they showed increased hair density by 90 ± 6 hair cm^2^ over initial values six months after treatment, which is significant. A higher concentration of IGF-1 and VEGF on injected site was also revealed. Additionally, an increased number of Ki67+ basal keratinocytes and improved vascularization were demonstrated. The delivery of the second PRP dose by conventional route enabled in vivo platelet activation and subsequent production of thromboxane, which activate additional platelets and amplify platelet aggregation.

Both patients displayed a marked increase in several observed hair parameters. The first patient demonstrated a significant gain in the frontal area as well. The achieved gain in hair parameters was more pronounced in the vertex region of both patients, which, in our view, is a consequence of the treatment plan. Namely, more injections were applied in the vertex area when compared to the frontal one, and the second patient was treated in the vertex area only. The analysis of the temporal area growth did not demonstrate a significant difference, which is a tribute to the quality of the assessment process. A slight increase in hair growth in the occipital area was noted, an occurrence that can be explained by the migration of a small quantity of PRP injected into the vertex. As this study with a candidate regimen for androgenetic alopecia was tested on 2 patients only, more studies are needed to confirm such effect. Therefore, the presented outcome should be supported by additional studies.

Our previous research shows positive effects in treating knee osteoarthritis using platelet-rich plasma (PRP) and autologous micro-fragmented adipose tissue with a stromal vascular fraction [12]. Autologous micro-fragmented adipose tissue contains mesenchymal stem cells (MSCs) that additionally have regenerative and immunomodulatory effects. Moreover, recent studies revealed gender-dependent differences in the stromal vascular fraction from lipoaspirate and micro-fragmented adipose tissue, which implies different therapeutic responses are needed in men and women [13]. These new findings indicate a potential beneficial effect of cell therapy in hair growth stimulation in addition to the currently tested PRP treatment. Further research on regenerative and immunomodulatory potential in androgenetic alopecia therapy will shed light on new possibilities in the treatment of the most common cause of hair loss.

Nowadays, when PRP is extensively used for hair growth stimulation, there is a great need to define uniform treatment regimens which address such conditions. In this direction, our pilot study encourages us to propose a combination of the Arthrex Angel System and a conventional collection of PRP. Injection technique used in our study is similar to techniques in previously published studies. Our comparative advantage is a reduced number of PRP administration time points and subsequently achieved significant result in hair growth.

## Figures and Tables

**Figure 1 jcm-11-07327-f001:**
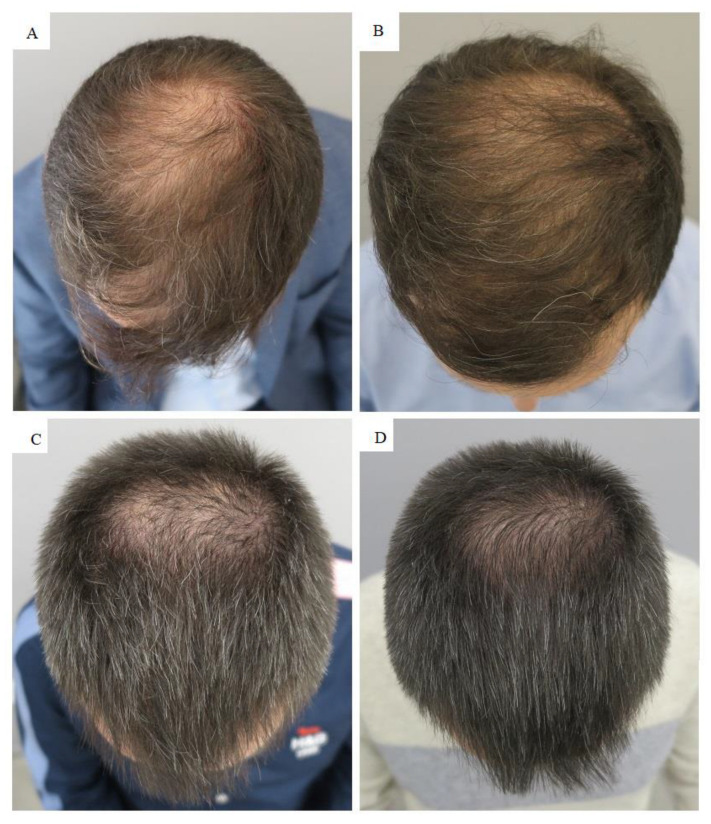
Results of autologous platelet-rich plasma (PRP) treatment in patients with androgenetic alopecia: patient 1 and 2 before (**A**,**C**) and six months after (**B**,**D**) treatment.

**Figure 2 jcm-11-07327-f002:**
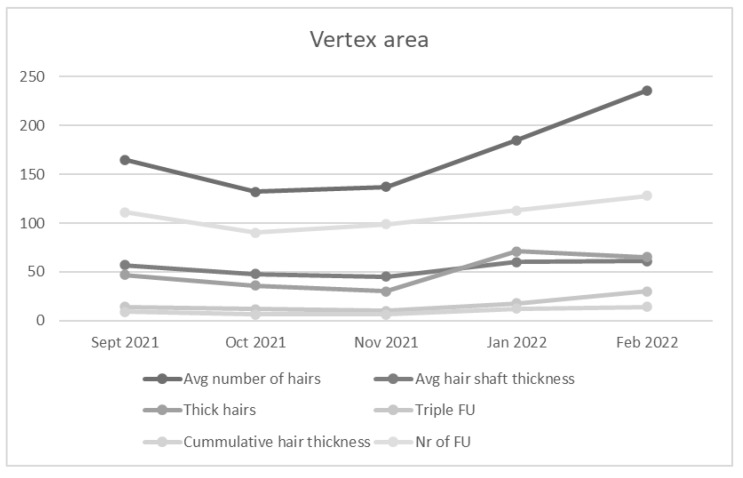
Average number of hairs, Average hair shaft thickness, Thick hairs, Triple FU, Cumulative hair thickness, and Number (Nr) of follicles (F) increased in the vertex area of the first patient.

**Figure 3 jcm-11-07327-f003:**
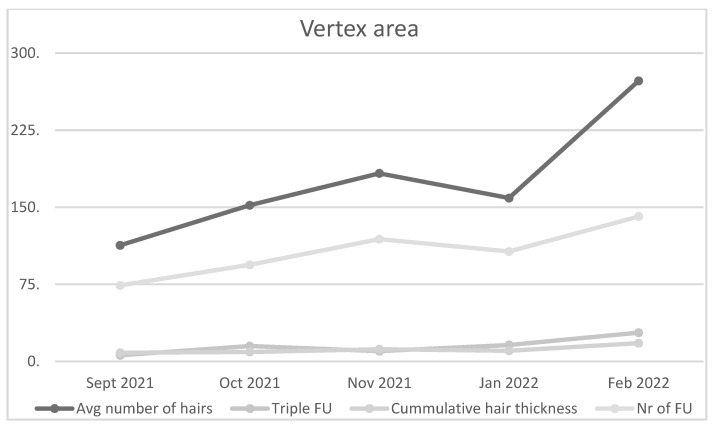
Average number of hairs, Triple FU, Cumulative hair thickness, and Number (Nr) of follicles (F) increased in the vertex area of the second patient.

**Figure 4 jcm-11-07327-f004:**
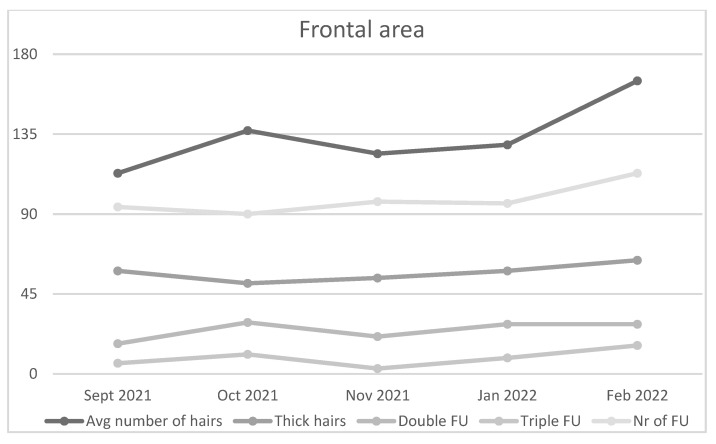
Average number of hairs, Thick hairs, Double and Triple FU, and Number (Nr) of follicles (F) increased in the frontal area of the first patient.

## Data Availability

Data used for analysis is contained within the article.

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
