# Peer review of "Autologous Platelet-Rich Plasma (PRP) for Treating Androgenetic Alopecia: A Novel Treatment Protocol Standardized on 2 Cases"

_jcm, 2022, doi:10.3390/jcm11247327_

Round 1

Reviewer 1 Report (Previous Reviewer 1)

The manuscript is much improved and is acceptable for publication.

Author Response

We thank this reviewer for recognizing our efforts to improve the quality of our manuscript.

Reviewer 2 Report (New Reviewer)

In this paper, the authors report two cases of androgenetic alopecia (AGA) treated with a novel PRP procedure.

Two cases are a poor series to determine the efficacy. More patients would give more effective results.

AGA is reported as "androgenic alopecia" in the introduction and the key word, then as "androgenetic" in the text. Please choose only one: androgenetic is mostly used.

Table 1 Hamilton-Norwood Scale can be omitted and described briefly in the discussion with a reference.

It would be recommended to describe the medical history of patients, especially if they underwent to other AGA treatment (minoxidil, finasteride, etc.) before PRP prcedure.

Author Response

We thank this reviewer for the suggestion which has significantly improved the quality of our revised manuscript.

We know that 2 cases are not enough for a general conclusion about PRP efficacy in this indication. Thus, we submitted the article as a case report and emphasized that further studies are needed for more confident conclusions (lines 213-216, discussion section).

According to your suggestion, we correct AGA to be the same across the manuscript – now we only use “androgenetic”. Each correction is marked using track changes.

In the first version of the manuscript, we did not incorporate Hamilton-Norwood Scale, and was only referenced in the literature. However, in the last major revision Reviewer 1 asked to incorporate such a table (“It would be useful to provide a table summarizing the Hamilton Norwood Scale.”). According to your suggestion, we will now omit the table, and reference is already included in the literature (number 9).

We already included a short statement about the medical history of our patients. Namely, both our patients are healthy and in generally good condition. Additionally, in the aspect of AGA, they did not receive any other medications or treatments. So far, it was written in lines 60-64: “Both patients were otherwise healthy and in good physical condition.” and “Patients did not receive any other treatment option for androgenic androgenetic alopecia before or during this study.”. 

Reviewer 3 Report (New Reviewer)

Platelet-rich plasma is a widely used treatment for the treatment of androgenetic alopecia, but the evidence that supports its use is not so extensive, so this type of study is very necessary. The study is well designed and written. However, it presents a low number of patients (only 2 cases). I encourage the authors to conduct a larger study.

Author Response

We thank this reviewer for constructive criticism and suggestions, which we will try to address in our reply.

As you said, although PRP is widely used for androgenetic alopecia treatment, standardized guidelines are not defined. Therefore, we did our best to contribute to the understanding of this treatment. We submitted our article as a case report because we have only two patients. In the discussion section, we clearly emphasized that further studies (on large patient series) are necessary. After informing the broad audience about this treatment option, we would do our best to test this treatment regimen on more patients in a structured study.

Round 2

Reviewer 2 Report (New Reviewer)

The authors have done little changes as suggested.

However, they present only two cases.  

This manuscript is a resubmission of an earlier submission. The following is a list of the peer review reports and author responses from that submission.

Round 1

Reviewer 1 Report

See attached

Reviewer 2 Report

it is an interesting case report. however, I found difficulty in understanding the methodology. I read it more than once to get the idea. so I would like to suggest that you have to simplify the methodology and make it clear and to the point . regarding the results, it contains detailed information. it would be better if you highlight the most important findings such as the highest and lowest instead of repeating all the data of the graph.